

# Pectoral muscle area index is an independent protective factor for mortality in sepsis patients: a retrospective observational study

Xin Li[1], Meijiao Li[2], Yongchang Sun[1] and Qingtao Zhou[1]

[1] Department of Respiratory and Critical Care Medicine, Peking University Third Hospital, Beijing, China
[2] Department of Radiology, Peking University Third Hospital, Beijing, China

## ABSTRACT

**Background.** Sepsis is an infection-induced systemic inflammatory response involving multiple mediators. Identifying risk factors for mortality in patients with sepsis is important for determining treatment strategies. Sarcopenia is a systemic pathology of the skeletal muscles associated with poor outcomes in patients with sepsis. However, there exists a gap in the literature regarding the thoracic muscle area and early outcomes of sepsis. Thus, this study investigated the relationship between 28-day survival and indicators of sarcopenia (pectoral muscle area and pectoral muscle density) from chest computed tomography images of patients with sepsis.

**Methods.** Patients ($n = 134$, median age $= 75$ years) who met the Sepsis-3 diagnosis criteria were included. Pectoral muscle area and density were measured in patients who underwent pulmonary computed tomography within $\pm 3$ days of admission. Univariate and multivariable factors associated with 28-day mortality were evaluated *via* Cox regression analysis. Factors with a single-factor $p < 0.05$ were included in the multivariate Cox regression analysis to evaluate the factors associated with 28-day mortality in sepsis.

**Results.** In this study, 88 patients survived, whereas 46 did not survive at the 28-day mark. Body mass index (BMI) ($P = 0.044$), pectoral muscle area ($P = 0.005$), pectoral muscle density ($P = 0.008$), and pectoral muscle area index ($P = 0.003$) significantly differed between patients who survived and those who did not survive. BMI was positively correlated with pectoral muscle area ($r = 0.516$, $P < 0.001$) and index ($r = 0.560$, $P < 0.001$). Acute Physiology and Chronic Health Evaluation II score (hazard ratio (HR) $= 1.119$, $P < 0.001$) constituted an independent risk factor for 28-day mortality, whereas the pectoral muscle area index (HR, 0.847; $P = 0.027$) was a protective factor for 28-day mortality in patients with sepsis. The pectoral muscle area index was associated with a reduced risk of early mortality in patients with sepsis.

Corresponding authors
Yongchang Sun, suny@bjmu.edu.cn
Qingtao Zhou, qtzhou75@163.com

## INTRODUCTION

Sepsis is an infection-induced systemic inflammatory response involving multiple mediators. Sepsis incidence rates have increased to 535 cases per 100,000 person-years and continue to rise (*Walkey, Lagu & Lindenauer, 2015*). Sepsis is a frequently encountered life-threatening condition in intensive care units (ICUs) worldwide and accounts for 12%–42.5% of ICU admissions (*Fleischmann-Struzek et al., 2020*). The pathogenic mechanism of sepsis is complex, and numerous studies and improvements in clinical procedures have accelerated its identification and treatment (*Rudd et al., 2020*). However, mortality rates remain at 34.7%–38.5% (*Bauer et al., 2020*), and the effects of sepsis are especially pronounced in older adults (*Lee et al., 2018*). Therefore, identifying risk factors for mortality in patients with sepsis is important not only for early risk stratification but also for future treatment strategies.

Sarcopenia is a progressive and systemic pathology of the skeletal muscles associated with frailty, functional impairment, and poor survival among older individuals (*Beaudart et al., 2017*; *Roberts, Collins & Rattray, 2021*). It is also associated with poor outcomes in chronic kidney disease (*Santana Gomes et al., 2022*), chronic obstructive pulmonary disease (*He et al., 2023*; *Dávalos-Yerovi et al., 2019*), sepsis (*Liu, Hu & Zhao, 2022*; *Cox et al., 2021*) and other diseases.

Most studies on sepsis and sarcopenia have analyzed patients with sepsis accompanied by intra-abdominal infections. The total cross-sectional area of the axial skeletal muscles (including the bilateral psoas, paraspinal, and abdominal wall muscles) at the level of the third lumbar vertebra (L3) has been analyzed using abdominal computed tomography (CT) (*Oh et al., 2022*; *Cox et al., 2021*; *Loosen et al., 2020*). *Cox et al. (2021)* found that acute muscle atrophy occurs in patients with sepsis caused by abdominal infection and persists for at least 3 months. Their study also revealed that baseline muscle loss, rather than acute muscle atrophy itself, is the primary predictor of poor prognosis and mortality in these patients. These findings suggest new prognostic indicators for sepsis in patients undergoing abdominal surgery.

As abdominal CT is routinely used to evaluate the extent and location of infection in patients with abdominal infections, it also enables the measurement of the L3 cross-sectional area. However, studies on the correlations between the thoracic muscle area and early outcomes of sepsis are scarce, and sepsis in the medical ICU (MICU) develops primarily from pulmonary infections (*Jouffroy et al., 2021*; *Abe et al., 2018*). With the onset of the COVID-19 pandemic in 2019, chest CT has gained increasing prominence. This development has enabled the utilization of chest CT for assessing the cross-sectional area of the pectoral muscles, allowing for the prediction of prognosis in patients with sepsis resulting from lung infections through the measurement of the pectoral muscle area (PMA). The PMA index (PMI) and PMA are indicators of sarcopenia that are associated with the duration of hospitalization and mortality in various diseases (*Ali & Kunugi, 2021*; *Diaz et al., 2018*).

This study investigated the correlation between PMA (including the pectoralis major and minor muscles) and pectoral muscle density (PMD) on chest CT images of patients with sepsis admitted to the MICU and their 28-day outcomes.

## METHODS

### Study design and patient sample

This retrospective study reviewed the clinical data of 134 patients admitted to the MICU between June 2015 and August 2019. Patients who were ≥18 years of age, diagnosed with sepsis based on the Sepsis-3 diagnostic criteria (*Singer et al., 2016*), and had an interval of ±3 days between chest CT imaging and admission were included in the study. The study excluded patients who were pregnant, who had human immunodeficiency virus or tuberculosis infection, who had a do-not-resuscitate or a discontinued tracheal intubation order, or who had an expected lifespan of <3 months owing to preexisting comorbidities, and who had sepsis requiring surgical treatment or caused by an infection related to a surgical procedure. Patients were assigned to survival or non-survival groups based on survival status at 28 days. The protocol was approved by the Institutional Review Board of Peking University Third Hospital (IRB No. M2019396). Written informed consent was obtained from the study participants.

### Clinical and laboratory data

The baseline characteristics of age, sex, weight, height, BMI, $PaO_2/FiO_2$, Acute Physiology and Chronic Health Evaluation (APACHE) II score, Sequential Organ Failure Assessment (SOFA) score, and duration of hospitalization were analyzed. Complete blood counts and levels of lactate, creatinine, blood urea nitrogen, and bilirubin were assessed on day 1 of admission.

### CT muscle imaging

PMA was measured using an inspiratory chest CT scan by two trained physicians who were unaware of the participant's data. The average of the two physicians' measurements was used as the final result. If the difference between the two physicians was large, a third physician was asked to re-evaluate. The analysts visually identified the superior aortic arch, followed by the first axial slice above the arch, as PMA. This location was selected because it was easily and consistently identified in all patients. The right and left pectoralis major and minor muscles were then identified in the anterior thorax, and their edges were manually segmented using predefined 250 and 90 HU attenuation ranges. The averages of two PMA measurements were analyzed. Differences in PMAs caused by height were corrected using the PMI, where the PMA in $cm^2$ was divided by the square of height in meters ($cm^2/m^2$).

### Statistical analysis

Data were statistically analyzed using SPSS, version 23.0 (IBM Corp., Armonk, NY, USA). The *P*-value threshold (<0.05) was applied to all analyses, including the regression. The Shapiro–Wilk test was used to evaluate the normality of the measurement data. Normally distributed data are expressed as mean ± standard deviation, non-normally distributed

data are expressed as median and interquartile range, and categorical data are presented as $n$ (%). Differences in means were assessed using Student's $t$-tests or analysis of variance, differences in medians were assessed using Mann–Whitney U tests, and differences in proportions were assessed using Chi-squared or Fisher's exact tests. Factors associated with 28-day mortality were examined using Cox proportional hazard regression models. Variables that were significantly associated with 28-day mortality were analyzed *via* univariate and multivariate Cox proportional risk regression analyses.

## RESULTS

### Baseline characteristics of patients

Figure 1 presents a flow diagram for the selection of 134 patients with sepsis. The median age was 75 years (59.5–84.0); 104 (77.6%) patients were male, and 112 (83.6%) patients had pulmonary infections. At 28 days, 88 patients survived, while 46 patients did not survive. The non-survival group had a significantly lower BMI than the survival group (23.3 $\pm$ 4.5 *vs.* 24.9 $\pm$ 4.2, $P = 0.044$). The lactic acid level (2.2 *vs.* 1.4, $P = 0.002$), SOFA score (8 *vs.* 6, $P = 0.023$), and APACHE II score (20 *vs.* 16, $P < 0.001$) were significantly higher in the non-survival group than in the survival group (Table 1). PMA (22.0 *vs.* 26.4, $P = 0.005$) and PMI (8.4 *vs.* 9.9, $P = 0.003$) were significantly lower in the non-survival group than in the survival group.

### Correlations between the PMA, PMI, and BMI

PMA, PMI, and BMI differed significantly between the groups (Table 1). BMI was positively correlated with PMA ($r = 0.516$, $P < 0.001$) and PMI ($r = 0.560$, $P < 0.001$; Fig. 2).

### Univariate and multivariable risk analysis of 28-day mortality in hospitalized patients with sepsis

As PMA correlated with height, PMI in the regression model analysis was used to exclude the effect of height. In both univariate analyses and multivariable Cox proportional hazards regression models, the APACHE II score was associated with an increased 28-day mortality risk in patients with sepsis (HR, 1.119; 95% confidence interval (CI) [1.059–1.182]; $P < 0.001$). In contrast, PMI was associated with a reduced 28-day mortality risk in patients with sepsis (HR, 0.847; 95% CI [0.732–0.981], $P = 0.027$; Table 2).

## DISCUSSION

This study included 134 patients with sepsis admitted to our institution's MICU. PMD, BMI, and PMA differed significantly between patients in the survival and non-survival groups. BMI was positively correlated with both PMA ($r = 0.516$, $P < 0.001$) and PMI ($r = 0.560$, $P < 0.001$). Although PMI is more difficult to determine than BMI, patients in MICUs are often admitted for pulmonary infection, and chest CT imaging is frequently performed because of the possibility of COVID-19 infection. Multivariable Cox proportional hazards regression analysis showed that PMI was more reliable than BMI for a 28-day prognosis of sepsis. Moreover, PMI was significantly associated with a reduced risk of 28-day mortality, whereas the APACHE II score was significantly associated with an increased risk of 28-day

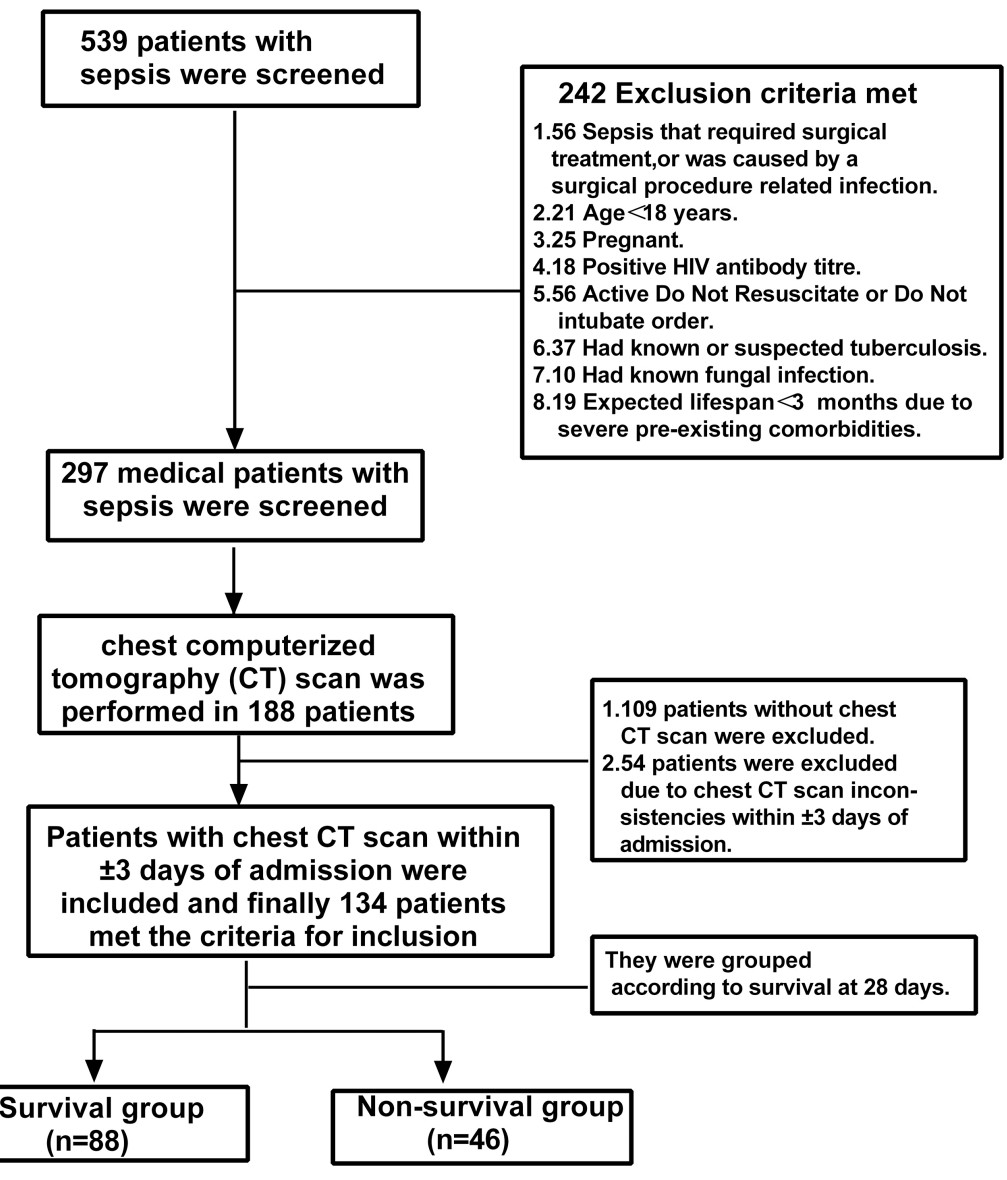

**Figure 1 Patient selection.** CT, computed tomography; HIV, human immunodeficiency virus.

mortality in patients with sepsis. BMI is easy to measure as the ratio of weight to height squared. It has previously been reported that BMI is an independent prognostic factor for 90-day survival in patients with medical sepsis and that patients with a lower BMI have a higher risk of death (*Zhou et al., 2018*). However, those studies had some limitations. The studies did not include patients with morbid obesity; the number of morbidly obese cases in China is relatively small, which renders the correlation trends of BMI and sepsis prognosis linear. In addition, the sources of infection differ between the MICU and the surgical ICU (*Martin-Loeches et al., 2019*), and the most common infection site in patients in the MICU is the lungs. Others have suggested that the lungs are the most frequent

**Table 1 Clinical and laboratory characteristics of the participants.**

| Variable | All patients (n = 134) | Survivors (n = 88) | Non-survivors (n = 46) | P-value |
|---|---|---|---|---|
| Age, years | 75.0 (59.5–84.0) | 72.0 (54.0–83.0) | 78.0 (71.8–84.0) | 0.020 |
| Male sex | 104 (77.6) | 67 (76.1) | 37 (80.4) | 0.571 |
| BMI[a], kg/m$^2$ | 24.3 ± 4.3 | 24.9 ± 4.2 | 23.3 ± 4.5 | 0.044 |
| Source of infections, n (%) | | | | |
| Lung | 112 (83.6) | 74 (84.1) | 38 (82.6) | 0.826 |
| Other infections | 22 (16.4) | 14 (15.9) | 8 (17.4) | 0.826 |
| Comorbidities, n (%) | | | | |
| COPD | 13 (9.7) | 11 (12.5) | 2 (4.4) | 0.228 |
| DM | 36 (26.9) | 24 (27.3) | 12 (26.1) | 0.883 |
| Hypertension | 63 (47.0) | 43 (48.9) | 20 (43.5) | 0.553 |
| Cerebrovascular disease | 28 (20.9) | 19 (21.6) | 9 (19.6) | 0.784 |
| Length of hospital stay (days) | 17 (10.5–25.0) | 18 (12.0–26.0) | 13 (7.0–21.5) | 0.023 |
| Laboratory data | | | | |
| Platelets, 10$^9$/L | 157.5 (102.0–220.5) | 166.5 (118.8–248.5) | 140.5 (83.8–191.3) | 0.060 |
| Lactic acid, mmol/L | 1.8 (1.2–2.7) | 1.4 (1.0–2.5) | 2.2 (1.5–3.5) | 0.002 |
| PaO$_2$/FiO$_2$, mmHg | 155.1 (100.0–225.8) | 163.0 (101.0–225.0) | 150.0 (93.5–229.0) | 0.168 |
| Creatinine, μmol/L | 98.0 (60.5–174.0) | 105.0 (66.0–188.0) | 83.5 (59.5–141.0) | 0.263 |
| TBIL, μmol/L | 15.7 (11.3–25.9) | 14.6 (9.4–22.6) | 17.9 (13.1–33.3) | 0.847 |
| BUN, mmol/L | 9.9 (6.6–15.6) | 9.2 (5.8–16.0) | 11.0 (7.3–15.4) | 0.596 |
| Albumin[a], g/L | 28.3 ± 4.8 | 28.5 ± 5.0 | 27.9 ± 4.4 | 0.499 |
| INR | 1.3 (1.1–1.4) | 1.2 (1.1–1.4) | 1.3 (1.2–1.6) | 0.279 |
| SOFA score | 7.0 (5.0–9.0) | 6.0 (4.0–9.0) | 8.0 (6.0–10.3) | 0.023 |
| APACHE II score | 16.0 (13.0–22.8) | 16.0 (12.0–20.0) | 20.0 (16.0–26.0) | <0.001 |
| Pectoral muscle area, cm$^2$ | 24.8 (19.3–33.6) | 26.4 (21.6–35.3) | 22.0 (15.7–29.0) | 0.005 |
| Pectoral muscle density, Hu | 35.9 (25.2–42.8) | 38.0 (30.2–44.0) | 33.8 (20.6–37.9) | 0.008 |
| PMI, cm$^2$/m$^2$ | 9.3 (6.7–11.5) | 9.9 (7.9–12.0) | 8.4 (5.7–10.2) | 0.003 |

**Notes.**

Values are expressed as n (%) or median (25%–75%) unless otherwise stated.

[a]Data was expressed as mean ± standard.

APACHE, Acute Physiology and Chronic Health Evaluation; BMI, body mass index; BUN, blood urea nitrogen; COPD, chronic obstructive pulmonary disease; DM, diabetes mellitus; FiO2, inspired fraction of oxygen; PaO2, partial pressure of oxygen; INR, international normalized ratio; PMI, pectoral muscle area index; SOFA, Sequential Organ Failure Assessment; TBIL, total bilirubin.

site of bacterial colonization. Sepsis is caused more frequently by lung infections than by abdominal infections (56.3% *vs.* 37.3%) and is more common among older individuals. Furthermore, mortality in the ICU and 1-year mortality in cases of sepsis caused by lung infection are related. Mortality rates in cases of sepsis caused by lung infections are higher than those in cases of sepsis caused by abdominal infections (31.7% *vs.* 12.6% and 45.4% *vs.* 24.4%, respectively) (*He et al., 2016*).

Patients with sarcopenia and sepsis may have overlapping risk factors, such as advanced age (*Rossi et al., 2018*). Sepsis is an acute disease caused by immune dysregulation and heightened inflammation, with high rates of morbidity and mortality. More than half of all sepsis cases occur in adults aged >65 years (*Rowe & McKoy, 2017*). Sarcopenia is also prevalent among adults aged >65 years and is associated with poor clinical outcomes,

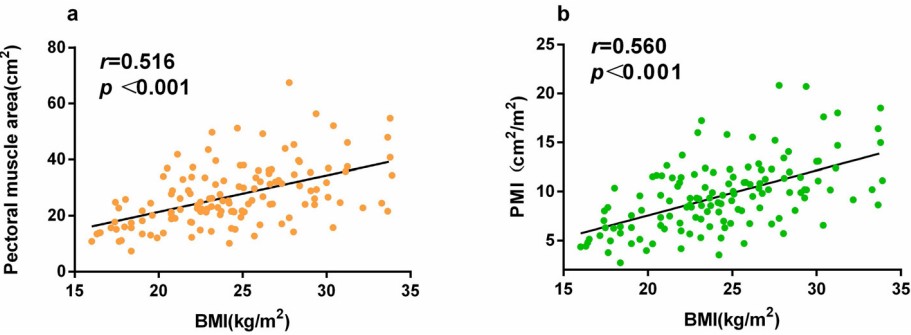

**Figure 2** (A) Correlation between BMI and PMA, and (B) correlation between BMI and PMI.

**Table 2** Cox regression analysis for determining risk factors for 28-day mortality among patients.

| Variable | Univariate HR (95% CI) | P-value | Multivariate HR (95% CI) | P-value |
|---|---|---|---|---|
| BMI, kg/m$^2$ | 0.911 (0.841–0.986) | 0.020 | | |
| PaO$_2$/FiO$_2$, mmHg | 0.994 (0.990–0.999) | 0.020 | | |
| SOFA score | 1.178 (1.076–1.289) | <0.001 | | |
| APACHE II score | 1.140 (1.086–1.197) | <0.001 | 1.119 (1.059–1.182) | <0.001 |
| Pectoral muscle index | 0.849 (0.761–0.946) | 0.002 | 0.847 (0.732–0.981) | 0.027 |

**Notes.**

HR, hazard ratio; CI, confidence interval; BMI, body mass index; PaO2/FiO2, partial pressure of oxygen in arterial blood to the fraction of inspired oxygen ratio; SOFA, Sequential Organ Failure Assessment; APACHE, Acute Physiology and Chronic Health Evaluation.

disability, and mortality (*Song et al., 2019*; *Trancă et al., 2017*). The interaction between sarcopenia at the time of hospital admission and sepsis exacerbates disease progression. Muscle loss is usually caused by a combination of reduced hormone anabolism and increased catabolic signaling mediated by pro-inflammatory cytokines, such as tumor necrosis factor-alpha (TNF-α) and interleukin 6 (IL-6). In particular, levels of TNF-α and IL-6 are elevated in the skeletal muscles of older patients (*Zanders et al., 2022*). In contrast, IL-6 levels are significantly elevated and function as a biomarker of outcomes in patients with sepsis (*Merz & Thurmond, 2020*; *Shamim, Hawley & Camera, 2018*). Interleukin-6 mediated sepsis-induced muscle atrophy through the gp130/JAK2/STAT3 pathway in a mouse model of sepsis, indicating the possibility of ICU-acquired weakness (*Lieffers et al., 2012*). Furthermore, skeletal muscles are essential for glucose conversion, protein synthesis (*Boutin et al., 2015*; *Prado & Heymsfield, 2014*), and the prevention of infection (*Ji et al., 2018*). This suggests that patients with sarcopenia may be more susceptible to new and exacerbated existing infections than those without sarcopenia, owing to a poorer immune status.

Peripheral muscle weakness should be quantified using Medical Research Council sum scores (MRC-SS), which include manual assessments of three functional muscle groups in the upper (shoulder abduction, elbow flexion, and wrist extension) and lower (hip flexion, knee extension, and foot dorsiflexion) extremities. Muscle strength is scored from

0 (no evident movement) to 5 (normal contraction against full resistance). However, from a clinical perspective, the physical examination of critically ill patients in the ICU is often hampered by issues such as preexisting neuromuscular disease, sedation, and delirium. These could be particularly relevant for sensory testing and MRC-SS assessment and could lead to increased interobserver variability. Therefore, skeletal muscle mass can be assessed using several radiological strategies, such as CT, ultrasound, bioimpedance analysis, and dual-energy X-ray absorptiometry (*Boutin et al., 2015*). Among these, CT is usually considered the gold standard for measuring body composition at the tissue or organ level (*Prado & Heymsfield, 2014*). CT has the following advantages regarding the measurement of the skeletal muscle area. First, CT has good repeatability. CT scanning conditions, such as voltage and layer thickness, are easily standardized, and results between different institutions or time points are highly comparable. Second, it has high resolution and accuracy. CT can accurately distinguish skeletal muscle and adipose tissue based on density and can integrate multiple layers of images. CT allows continuous tomography to support 3D reconstruction, not only measuring the area of a single cross-section but also calculating muscle volume for a more comprehensive assessment of overall or local muscle mass. In addition, CT has a wide range of applications in patients with lung infections. In particular, after the novel coronavirus epidemic, the use of chest CT has significantly increased. Finally, CT can be performed within a short period, as a single scan takes only a few seconds or minutes, which is more convenient for severely ill patients or those who cannot cooperate for a long time. However, this study's findings differ from those of previous studies in that it assessed whether PMA and pectoral muscle mass measured on chest CT images are associated with 28-day mortality in patients with sepsis in the MICU. As noted above, patients with sepsis and abdominal infections have been the focus of many studies; thus, the psoas major muscle area has been assessed using abdominal CT. In contrast, the predominant cause of sepsis in patients in the MICU is pulmonary infection; thus, it is more accessible to use chest CT to evaluate skeletal muscle indices.

In addition, a statistically significant age difference was found between the survival and non-survival groups in this study (72 *vs.* 78 years, $P = 0.020$). However, in a subsequent multivariate review, it was found that age was not an independent factor for 28-day mortality from sepsis. The reasons for this are as follows. First, early identification and treatment improved the short-term prognosis of sepsis (*Rudd et al., 2020*), and the death time of patients evaluated at 28 days was shorter. In this study, the average length of hospitalization was 17 days; that is, the time of death of patients evaluated 11 days after discharge was stable. In a recent study of 2,322 patients with sepsis, 90-day mortality was associated with patient age (*Xie et al., 2020*). In addition, recent studies have found that the death pattern related to sepsis has changed to a three-phase pattern (*Delano & Ward, 2016*). The early peak mortality gradually decreased due to early identification and active treatment, but the number of patients with sepsis who died at the third peak gradually increased, and the peak occurred 60–90 days after sepsis (*Ramoni et al., 2024*). Therefore, in this study, although there was a difference in age between the survival and death groups, age was not an independent influencing factor for 28-day mortality from sepsis.

This study has certain limitations that warrant acknowledgment. Firstly, given that the median age of patients in the MICU is 75 years, the findings of this article primarily pertain to elderly patients with sepsis. To enhance generalizability, future research should focus on conducting multicenter studies to ensure that the results are representative of a broader population. Second, this study only considered the MICU, and patients with mostly lung infections may not be representative of surgical and other ICUs. However, because patients in the MICU mainly have lung infections, chest CT is more widely used because it allows better observation of the chest muscle area. Third, the sample size of this study was small, which may have caused bias. Although our sample size is relatively limited, after calculation based on the Events per Variable (EPV), it can be concluded that our sample size is sufficient to validate this hypothesis. For the clinical prediction model, we adopted the empirical and the 20 EPV method to calculate the sample size. Ultimately, five predictive parameters were included in the review equation, with a minimum required sample size of 100 cases. Finally, this study was retrospective; therefore, future prospective studies are necessary to verify our findings.

## CONCLUSIONS

PMI was identified as an independent protective factor against 28-day all-cause mortality in patients with sepsis in the MICU.

## ACKNOWLEDGEMENTS

We thank all those who assisted in conducting and publishing this study.

### Funding
This study was funded by the National Natural Science Foundation of China (No. 82272197) and the clinical cohort construction program of Peking University Third Hospital (No. BYSYDL2021019). The funders had no role in study design, data collection and analysis, decision to publish, or preparation of the manuscript.

### Grant Disclosures
The following grant information was disclosed by the authors:
National Natural Science Foundation of China: 82272197.
Peking University Third Hospital: BYSYDL2021019.

### Competing Interests
The authors declare there are no competing interests.

### Author Contributions
- Xin Li performed the experiments, analyzed the data, prepared figures and/or tables, and approved the final draft.

- Meijiao Li performed the experiments, prepared figures and/or tables, and approved the final draft.
- Yongchang Sun conceived and designed the experiments, authored or reviewed drafts of the article, and approved the final draft.
- Qingtao Zhou conceived and designed the experiments, analyzed the data, authored or reviewed drafts of the article, and approved the final draft.

## Human Ethics

The following information was supplied relating to ethical approvals (i.e., approving body and any reference numbers):

The Institutional Review Board of Peking University Third Hospital (IRB, M2019396) approved the protocol of this retrospective study of 134 patients admitted to the MICU between June 2015 and August 2019.

## Data Availability

The raw data are available in the Supplemental files.

## Supplemental Information

Supplemental information for this article can be found online at http://dx.doi.org/10.7717/peerj.19689#supplemental-information.

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
