# Peer review of "Pectoral muscle area index is an independent protective factor for mortality in sepsis patients: a retrospective observational study"

_PeerJ, doi:10.7717/peerj.19689_

## Round 0.1 · original submission · Major Revisions

· Academic Editor

Major Revisions

I have completed my evaluation of your manuscript. The reviewers recommend reconsideration of your manuscript following major revision. I invite you to resubmit your manuscript after addressing the comments below. When revising your manuscript, please consider all issues mentioned in the reviewers' comments carefully: please outline every change made in response to their comments and provide suitable rebuttals for any comments not addressed. Please note that your revised submission may need to be re-reviewed.

Reviewer 1 ·

Basic reporting

no comments

Experimental design

no comments

Validity of the findings

no comments

Additional comments

The authors of the manuscript have shown that the Pectoral muscle area index can be utilized for early sepsis diagnosis. The abstract is structured well and focuses on the content of the manuscript.
There is a lack of detailed explanations to justify the study. Lines 62-63 require further detailing.
To further validate the study can you show Pearsons’s correlation of PMI /PMA with SOFA score or APACHE II?
In the conclusion, the author should mention the clinical application of this finding.
Correct the figure legends
Figure 2: Correlations between BMI and the PMA (a) and PMI (b)
It shows Correlations between BMI and the PMA (a) not (b)
Instead, figure b is showing a correlation between BMI and PMI
In addition, the manuscript is written in professional, unambiguous language. With these revisions, the manuscript should be acceptable.

·

Basic reporting

Dear authors,
Thank you for the opportunity to review your manuscript titled: “Pectoral muscle area index is an independent protective factor for early prognosis of sepsis: a retrospective observational study”. Your research is very relevant to the field and fits the scope of the journal. Although there is potential in the manuscript, a major revision is required.
According to the editorial criteria:
MAJOR CONCERNS
- Raw data file: The variables indicating admission and discharge time should be removed from the raw data file to ensure confidentiality of the participants.
- Ethical approval: An officially signed English translation of the ethical approval form is missing. The provided English forms seem to be adjusted after signing by use of either a PDF-editing program or an AI translation tool.
- Analysis: The rationale for excluding PMA in the multivariate analysis seems to be lacking and I would suggest the authors include their rationale for this in the response to this revision or re-do the regression analysis with PMA as a factor.
- Discussion: The discussion seems to lack some depth and connection with the aims of the study. I would suggest to the authors that they follow the structure of the primary and secondary aims of the study to describe the most prominent results first. Also, the extensive discussion of the relationship between BMI and mortality is interesting, but not within the scope of this study. The discussion therefore needs some major restructuring and reflection on relevant literature, in which human-subject studies are preferred.
BASIC REPORTING
Title: The authors use ‘for early prognosis of sepsis’ in their title. I would suggest changing the title to ‘Pectoral muscle area index is an independent protective factor for mortality in sepsis patients’, or something similar, to adhere to the study focus and findings.
Line 24. Remove “cytokines” in both the abstract and introduction, as this study is not in the field of immunology. Use “multiple mediators” instead for a broader and more accurate description.
Line 28. Replace “correlation” with “relation” in the abstract. Mentioning “correlation” may detract from the depth of the study, as the multivariate analysis is already a stronger statistical approach.
Line 30. Correct the sample size to n=134 instead of 135 in the abstract for consistency with the study.
Line 31. To save space omit the phrase “in this study” where possible.
Line 33. Also mention how the univariate analyses were conducted here (between group analyses and correlation testing)
Line 36. add (BMI), to replace later in line 38, and below in the rest of the paper.
Line 50. It would be a valuable addition to the rationale for this paper to provide some information on how many patients suffer from sepsis each year in the ICU.
Line 62. This sentence requires some explanation. What does this add to the introduction? Why is it relevant that most studies have included sepsis accompanied by intra-abdominal infections?
Line 63 – 71. Although the scope of the introduction is clear, I believe some restructuring is necessary. Please look closely at how the separate subjects in the introduction relate to each other. Using white spacing or connecting/concluding sentences can provide clarity.
Line 108. Specify that the p-value threshold (<0.05) applies to all analyses, including regression.
Line 137. The description of “Hazard Ratio (HR)” has already been introduced in the abstract.
Discussion overall. Please refrain from repeating parts of the introduction or the result section in the discussion. State the most important findings in the first paragraph and then reflect on this with relevant literature.
Line 148-155. This part can be removed or significantly shortened as it is beyond the scope of this study.
Line 165-169. This part should be mentioned earlier in the discussion as it reflects on the main finding of the study.
Line 170-177. This part seems to be quite similar to parts of the introduction and should be removed.
Line 179-180. This is the main finding of the paper and the authors should give it a more prominent place in the discussion.
Line 181-185. This seems to be very repetitive as it is also mentioned in the discussion or should be more relevant in the introduction.
Line 199-203. What do the authors want to say with this part? How is this relevant in light of the scope of the study and the main findings?
Line 207-208. It would be relevant to expand a bit more on the place of CT compared to these other measurement tools.
Line 215. Do the authors mean it is more appropriate or more accessible?
Line 217. The age difference in this study does not seem to be a real limitation, as age was included in the regression analysis. It would be an interesting finding to also discuss in the results and discussion.
Line 219. This could be tested by introducing an interaction term to the model using age * sepsis.
Line 223. It would be relevant to add a retrospective sample size calculation or some sort of explanation on this in the manuscript. It is not clear to the reader if the sample size was appropriate at this moment.
Table 2. Please write the abbreviations in alphabetical order here.
Figure 1. The quality of this figure seems to be low. Please add a version of higher quality.
Figure 2. My suggestion would be to remove this figure from the paper. The correlation between BMI and PMA/PMI is beyond the scope of this paper.
Literature References
- Line 51. Reference Fleischmann et al., 2016. Ideally, more recent references should be included, for example: Fleischmann-Struzek C, Mellhammar L, Rose N, Cassini A, Rudd KE, Schlattmann P, Allegranzi B, Reinhart K. Incidence and mortality of hospital- and ICU-treated sepsis: results from an updated and expanded systematic review and meta-analysis. Intensive Care Med. 2020 Aug;46(8):1552-1562. doi: 10.1007/s00134-020-06151-x. Epub 2020 Jun 22. PMID: 32572531; PMCID: PMC7381468.
- Line 53. Reference: (Tiru et al., 2015). This reference appears to focus more on economic evaluation rather than supporting the specific statement. Moreover, there is more recent literature available on the pathogenic mechanisms of sepsis, its clinical identification, and treatment.
- Line 55. Reference: (Martin, Mannino & Moss, 2006). There is more recent literature available regarding sepsis in elderly patients.
- Line 58. Reference: (Mitchell et al., 2012). Line 61. Reference: (Vestbo et al., 2006). Line 156. Reference: (Sayer et al., 2013). Line 176. Reference: (Mayr et al., 2010). Line 184. Reference: (Ryall, Schertzer & Lynch, 2008). More recent literature is available.

Experimental design

EXPERIMENTAL DESIGN
Line 32. How did you determine that three days is the timeframe for measuring PMA and PMD, considering that three days is a wide window that allows room for bias regarding the timing of patient admission and the tomography record? Especially when adjusting for BMI, as metabolic demands during those three days could cause changes in this value relative to admission.
Line 38. The rationale for testing the correlation between BMI and PMA/PMI is unclear to me in this study, as it does not align with the research questions. Could the authors either remove this or provide further explanation for this?
Line 54. Interval 40% - 60%. Bauer et al. reported that "mortality in sepsis and septic shock shows an average 30-day mortality of 34.7% (95% CI 32.6–36.9%) and a 90-day mortality in septic shock of 38.5% (95% CI 35.4–41.5%)." Since these data are sensitive, the 60% figure appears to be an overestimation and should be revised for greater accuracy.
Line 79. What is the rationale for including patients until 2019? Would it be possible to extend the patient cohort to more recent?
Line 80. What do the authors mean by +/- 3 days? What is the range of inclusion date? And what is the impact of this ‘later’ assessment of the CT data? This should be mentioned in the limitation section of the discussion.
Line 91. What is the rationale for including data on the oxygenation index in this paper?
Line 97. What do the authors know about the intra-observer variability in their study? Did you use a second assessor in this case and how many analysts were involved?
Line 97-104. It is a very nice addition of the authors to clearly explain the study measurement procedures, like the CT muscle imaging.
Line 106–115:
- While it provides detailed descriptions of how variables will be analysed according to their nature, it lacks details on how data behaviour was assessed (e.g., use of Shapiro-Wilk test or visual inspection of histograms).
- It also does not mention how the proportional hazards assumption was evaluated (e.g., using Schoenfeld residuals and/or visual inspection of log-minus-log plots).
- What criteria were used to decide whether continuous variables would be included directly, transformed, or categorized in the Cox model?
- Please also mention if and how correlations were separately tested in this study.
Line 133. Please explain which factors/confounders were included in the regression model and how the authors decided on this.
Line 192–194: Avoid including evidence from animal models unless explicitly relevant to this study. This inclusion seems disconnected from the rest of the research.

Validity of the findings

VALIDITY OF THE FINDINGS
Line 134. It is unclear to me why the correlation with height resulted in the exclusion of PMA in the regression model analysis. Could the authors elaborate on this? If BMI was already in the model, it may also be unnecessary to include height? As the main goal of the study is to assess both PMA and PMI in relation to mortality, removing PMA this early in the analysis has a great impact on the conclusions that can be drawn from the results of this paper.
Line 145-146. This sentence reads like PMA was not associated with mortality, but the authors did not include this factor in the analysis.

Additional comments

GENERAL COMMENTS
The research is relevant and fills a knowledge gap in the early prognosis of sepsis by introducing the pectoral muscle area index as a protective factor. However, the justifications for methodological choices (the 3-day interval for PMA measurements, not including PMA) need to be strengthened to avoid potential bias. Besides, statistical methods, especially handling of continuous variables and the evaluation of proportional hazards, need better documentation for transparency and replicability.
The authors should update some references with more recent literature to align with current knowledge.

·

Basic reporting

Thank you for your efforts
1- It is not clear why the study considered the Pectoral muscle area index as independent protective
factor for early prognosis of sepsis ?
2-How did the authors calculate sample size ?
3-how did the study overcome the potential bias ,
4-study did not look at gender differences in muscle between male and female
5-How the original weight did affect the results
6- Tables did not show the used tests

Experimental design

Need to address number 2 and 3 in the reports

Validity of the findings

The study need to clarify why the Pectoral muscle area index was an independent variable , and to classify the other studied variables , SOFA ,APATCHI , Lactic Acids

The justification of sample size has to be explained and clarify the level of confidence ,error margin , power ,
end point

Additional comments

no more comments

---

## Round 0.2 · Minor Revisions

· Academic Editor

Minor Revisions

I have completed my evaluation of your manuscript. The reviewers recommend reconsideration of your manuscript following minor revision. I invite you to resubmit your manuscript after addressing the comments below. When revising your manuscript, please consider all issues mentioned in the reviewers' comments carefully: please outline every change made in response to their comments and provide suitable rebuttals for any comments not addressed. Please note that your revised submission may need to be re-reviewed.

**Language Note:** The review process has identified that the English language must be improved. PeerJ can provide language editing services - please contact us at [email protected] for pricing (be sure to provide your manuscript number and title). Alternatively, you should make your own arrangements to improve the language quality and provide details in your response letter. – PeerJ Staff

Reviewer 1 ·

Basic reporting

The authors have addressed the queries and improved

Experimental design

The authors have addressed the queries and improved

Validity of the findings

The authors have addressed the queries and improved

Additional comments

The authors have addressed the queries and improved

·

Basic reporting

* Overall: I thank the authors for their thorough response to the comments of the reviewers. Although the manuscript improved significantly, there are some minor revisions to consider.

* English language: The authors attempt to share a lot of information in their manuscript, including some physiological aspects in the discussion. As the paper is quite dense in this information, it remains a tough read. I would recommend the authors to include a professional English writing check in their revisions for the final version, to improve readability.

* Literature references and background: The authors have improved and updated the literature reference as requested.

* Professional article structure, figures and tables + raw data: The structure of the article is sufficient, although some improvements can be made to the discussion section by removing parts that are more suitable for the introduction or methods section. In addition, the level of depth in the reflection, including the more physiological pathway descriptions, can distract from the main message of the manuscript. The raw data file has been updated based on the reviewer comments.

* Relevant results to hypotheses: This has improved significantly from the previous version.

Experimental design

* Scope of the research: The scope of the research falls within the aims and scope of the journal

* Research question: Although the research question is well defined and the authors provided clarification on the relevance of the study, I believe that this study still does not completely align with current research gaps.

* Technical & ethical standards: The authors have clarified some lacking information regarding the methodology in their response. In addition, the newly provided English ethical approval form seems legitimate.

* Methods: The methods are described in sufficient detail and information. It would be helpful if the authors provided the information regarding the sample size to the official manuscript, as they managed to adhere to this predefined goal and the limitation of a small sample size may not be appropriate.

Validity of the findings

* Underlying data: The provided underlying data is sufficient to support the statements in this manuscript

* Conclusions: The conclusions are well stated and link adequately to the research question and results.

Additional comments

In general, the manuscript improved significantly. Thorough English language editing will likely improve the readability of the paper.

·

Basic reporting

This is revised manuscript ,Pectoral muscle area index is an independent protective factor for mortality in sepsis patients : a retrospective observational study

The authors responded adequately to my comments , I have no more further comments

Experimental design

Adequate

Validity of the findings

Adequate

Additional comments

Accept

---

## Round 0.3 · accepted · Accept

· Academic Editor

Accept

Dear Author,
It is a pleasure to accept your manuscript entitled "Pectoral muscle area index is an independent protective factor for mortality in sepsis patients: a retrospective observational study" in its current form for publication in PeerJ.